

**Animal burrowing at cold seep ecotones boosts productivity by linking macromolecule turnover with chemosynthesis and nutrient cycling**

Maxim Rubin-Blum[1,2*], Eyal Rahav[1], Guy Sisma-Ventura[1], Yana Yudkovski[1], Zoya Harbuzov[1], Or Bialik[2,3], Oded Ezra[2], Anneleen Foubert[4], Barak Herut[1,2], Yizhaq Makovsky[2]

[1]Israel Oceanographic and Limnological Research, National Institute of Oceanography, Haifa, Israel

[2]Charney School of Marine Sciences (CSMS), University of Haifa, Haifa, Israel

[3]Institute of Geology and Palaeontology, University of Muenster, Münster, Germany

[4]Department of Geosciences, University of Fribourg, Fribourg, Switzerland

[*]Correspondence: mrubin@ocean.org.il





**Abstract**

Hydrocarbon seepage at the deep seafloor fuels flourishing chemosynthetic communities. These seeps impact the functionality of the benthic ecosystem beyond hotspots of gas emission, altering the abundance, diversity and activity of microbiota and fauna, and affecting geochemical processes. Yet, these chemosynthetic ecotones (chemotones) are far less explored than the foci of seepage. To better understand the functionality of chemotones, we: i) mapped seabed morphology at the periphery of gas seeps in the deep Eastern Mediterranean Sea, using video analyses and synthetic aperture sonar; ii) sampled chemotone sediments and described burrowing using computerized tomography; iii) explored nutrient concentrations; iv) quantified microbial abundance, activity and $N_2$ fixation rates in selected samples and v) extracted DNA and explored microbial diversity and function using amplicon sequencing and metagenomics. Our results indicate that the gas seepage yields gradients of burrowing intensity at the seep ecotones, especially by the ghost shrimp *Calliax lobata*. This burrowing alters nitrogen and sulfur cycling through the activity of diverse microbes. Burrow walls form a unique habitat, where macromolecules are degraded by Bacterioida, and their fermentation products fuel sulfate reduction by Desulfobacterota and Nitrospirota. These in turn support chemosynthetic Campylobacterota and giant sulfur bacteria *Thiomargarita*, which can aid *C. lobata* nutrition. These interactions may support enhanced productivity at seep ecotones.

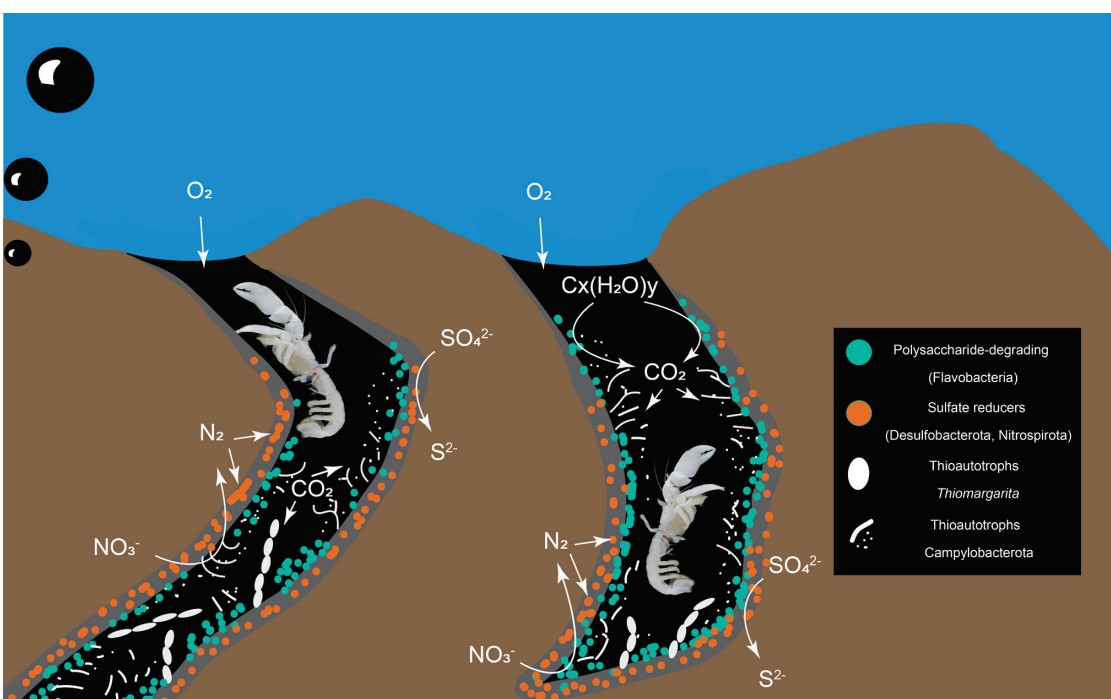



## 1. Introduction

Most organisms in the deep-sea biosphere thrive under extreme energy limitation (Orsi, 2018). In this dark, energy-limited environment, the natural discharge of fossil hydrocarbons results in accelerated biogeochemical dynamics, creating unique geobiological habitats - cold seeps (Joye, 2020). Cold seeps form the richest and most productive ecosystems that are widespread across continental margins (Joye, 2020; Levin and Sibuet, 2012). The transition zones between these highly productive ecosystems and the often impoverished deep-sea benthic habitats are considered chemosynthetic ecotones, that is, chemotones (Ashford et al., 2021). The chemotone communities benefit from chemosynthetic production, which can deliver food supplements through the water column, and intrinsic chemosynthetic producers in the sediments, resulting in intermediate biomass, richness and distinct species composition (Ashford et al., 2021). These areas of interactions and transition affect the deep-sea ecosystem services, most importantly, carbon cycling and sequestration, as well as fisheries production (Levin et al., 2016). However, our knowledge of chemotone communities is limited, and their functionality has not been studied in detail, e.g. (Åström et al., 2022; Amon et al., 2017; Ritt et al., 2011; Ashford et al., 2021).

Chemical and temperature clines around seep edifices may determine the biogeochemical processes at chemotones (Levin et al., 2016). In turn, surface water productivity and fluxes of photosynthetic carbon to the seep area may also impact the extent of seep chemotones (Ashford et al., 2021). For example, the strong pelagic-benthic coupling is limiting the chemosynthetic inputs in some Arctic seeps (Åström et al., 2022). Cold seeps are also pervasive in the warm (13.5 and 15.5 °C) deep waters of the ultraoligotrophic Southeastern Mediterranean Sea (SEMS) (Olu-Le Roy et al., 2004; Kormas and Meziti, 2020; Bayon et al., 2009; Lawal et al., 2023; Herut et al., 2022; Coleman et al., 2011; Coleman and Ballard, 2001). In high contrast to the Arctic seeps, the weak benthic-pelagic coupling in the ultraoligotrophic SEMS may lead to sharp gradients between the active seep sites and the energy-limited deep benthos (Sisma-Ventura et al., 2022). The warm deep-sea temperatures may alter the availability of dissolved gases and boost microbial activity in SEMS, potentially affecting the biogeochemical gradients at seep clines (Rahav et al., 2019; Mavropoulou et al., 2020; Roether and Well, 2001).

Here we focus on cold seep chemotones at the toe region of Palmahim Disturbance, an evaporite rooted, progressive, large-scale (~15 x 50 km), rotational submarine slide on the SEMS continental margin, offshore Israel (Garfunkel et al., 1979; Gadol et al., 2020); (Fig. 1). These seeps are characterized by local hotspots of slow diffusive discharge, where authigenic carbonates are formed (Rubin-Blum et al., 2014a, b; Weidlich et al., 2023) with dispersed bivalve beds, indicative of substantial past, and limited present activity (Beccari et al., 2020). Yet, hotspots of marked discharge of gas-rich brines, forming small (<40 m wide) brine pools, were recently discovered (Herut et al., 2022). The sediments at Palmahim Disturbance seep chemotones do not host the typical seep fauna but are inhabited by dense populations of a burrowing ghost shrimp, *Calliax lobata*, which alter the morphology of the sediment surface for tens of meters away from the seepage hotspots (Makovsky et al., 2017; Beccari et al., 2020; Basso et al., 2020). Similar burrowing is prominent in the periphery of SEMS seeps, as it was, for example, observed in the hemipelagic sediments near the Amon mud volcano at the Nile Deep Sea Fan (Ritt et al., 2011; Blouet et al., 2021). At Palmahim Disturbance, the sediments in the seep periphery are characterized by elevated nutrient fluxes and oxygen consumption, as well as increased microbial activity compared to the background stations (Sisma-Ventura et al., 2022).



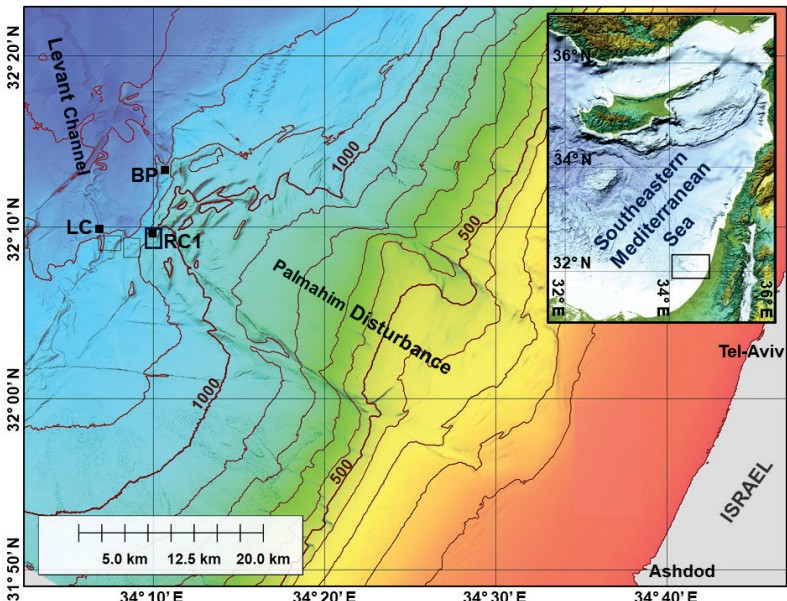

Figure 1. A bathymetric map of Palmahim Disturbance and its vicinity (color-coded and contoured with a 100 m spacing). This study is focused on seafloor gas seeps in the ridge-crest pockmark system (RC1) at the southwestern edge of the Disturbance, the proximate part of the Levant Channel (LC), and the brine pool (BP) area to the north. Black dots mark the sites of box-core sampling, and the black frames outline the ROV surveying analysis maps of Figure 2D-E.

Thus, microbes likely play a key role not only in hotspots of seepage but also in the seep ecotones. However, to date, most studies of seep microbiota focus on seepage hotspots. At these hotspots, seep ecosystems support diverse assemblages of microorganisms, with cosmopolitan distribution and key biogeochemical functions (Ruff et al., 2015; Teske and Carvalho, 2020). At the seepage hotspots, microbes catalyze key processes such as anaerobic oxidation of methane (AOM), sulfate reduction and chemosynthesis (Knittel and Boetius, 2009; Boetius and Wenzhöfer, 2013; Teske and Carvalho, 2020). The marked primary productivity at seeps may boost secondary microbial productivity, via, for example, the fermentation of organic macromolecules (Zhang et al., 2023). Such microbial communities are characteristic of Palmahim Disturbance seepage hotspots (Rubin-Blum et al., 2014b). Yet, little is known about the diversity and function of microbial communities that occupy the seep periphery, in the SEMS seeps in particular.

Here, we aimed to understand the extent and functionality of the burrowed sediments at Palmahim Disturbance seep chemotones. We thus mapped the bioturbation using analyses of video collected with remotely operated vehicles (ROVs) and high-resolution acoustic mapping with synthetic aperture sonar (SAS), mounted on an autonomous underwater vehicle (AUV). Hypothesizing that the bioturbated sediments at Palmahim Disturbance seep chemotones host unique microbial communities, which may function synergistically to exploit limited fluxes of seep carbon, we investigated their diversity and function using metagenomics analyses of samples collected at the seep periphery (Sisma-Ventura et al., 2022). We investigated the activity of chemotone microbes using the leucine uptake assay and evaluated their ability to fix dinitrogen. Previous studies suggest that ghost shrimps (Axiidea), garden and groom their burrows by incorporating organic matter into the walls through mucus excretions, creating microniches within the burrow walls (Coelho et al., 2000; Astall et al.,




1997; Dworschak et al., 2006; Abed-Navandi et al., 2005; Gilbertson et al., 2012; Laverock et al., 2010; Papaspyrou et al., 2005). In seeps, burrowing may enhance carbonate precipitation in areas with very low methane flux and/or diffusive seepage (Weidlich et al., 2023). We thus hypothesized that the burrowing activity of *C. lobata* can alter the diversity and functionality of these microbes, introducing new metabolic adaptations and handoffs. Therefore, we focused on the specific sediment layers where burrows were present.

## 2. Methods

### 2.1 Exploration of the Palmahim Disturbance gas seeps

Palmahim Disturbance toe region comprises a set of hundreds of meters high and ~1 km wide fold ridges and is bounded to the west by the ~500 m wide and ~40 m deep easternmost channel of the deep sea fan of the Nile River, the Levant channel (Gvirtzman et al., 2015). We investigated three seepage domains: i) Hundreds of meters wide pockmark systems

at the crest of three ridges marking the southwest of the Palmahim Disturbance toe, with a particularly large pockmark at the crest of the eastern ridge (RC1; **Figure 1**); ii) Elongate pockmarks along the sides of the Levant Channel (LC; **Figure 1**); iii) Brine pools at the northern part of Palmahim Disturbance toe region (BP; **Figure 1**). We collected sediment samples and survey data during several cruises to the Palmahim Disturbance offshore Israel, comprising: i) The E/V Nautilus cruise in 2011 (Coleman et al., 2012); ii) EUROFLEETS2-SEMSEEP cruise in the framework of the EUROFLEETS2 expeditions

in 2016, onboard R/V AEGAEO (Makovsky et al., 2017; Basso et al., 2020) and iii) R/V Bat-Galim cruises in 2020-23 (Sisma-Ventura et al., 2022; Herut et al., 2022; Rubin-Blum et al., 2024). In the latter cruises, sediment collection was supplemented with an AUV-based seafloor survey using a synthetic aperture sonar (SAS, Kraken Robotics Inc. MinSAS-120), providing a backscatter image of the seafloor with a constant ultra-HD 3x3 cm resolution across a 120 m wide swath along each side of the AUV track. The SAS data were collected in the framework of a 70 km long reconnaissance survey

that traversed the Palmahim Disturbance toe region. SAS images collected in some specific areas were manually correlated with the relevant ROV videos by identifying specific seafloor morphologies.

### 2.2 Burrowing quantification

The HD video data of three consecutive dives of the EV Nautilus 2011 cruise to the three ridge-crest pockmarks in the southwestern part of Palmahim Disturbance toe (RC1, **Figure 1**) were systematically analyzed in an ArcGIS desktop (Esri).

The intensity of burrowing was semi-quantified by classifying to three levels the density of burrowing mounds observed on the seafloor in the field of view of the ROV camera during surveying, and by SAS (**Figure 2, A-D**): i) Low – where the seafloor is generally smooth with the possible presence of a few sporadic burrowing mounds; ii) Medium – when multiple burrowing mounds are continuously viewed; and iii) High – where burrowing mounds are pervasive across the majority of the field of view. Analysis mapped markers of active seepage, including bubbles discharge, bacterial mats, chemosynthetic

fauna, and seafloor burrowing. The results were combined to generate a maps of seepage indicator distribution (**Figure 2, E**).



### 2.3 Sediment sample collection

Sediments were sampled using box-corers, targeted with an ultra-short baseline (USBL) positioning system in the large ridge-crest pockmark RC1 (AG16-17BC1 (2016), Box1 and 2 (2020), all within ~20 m positioning accuracy of 32° 09.60' N, 34° 10.00' E, water depth circa 1035 m) (**Figure 1, Table 1**). Additional Flare 1 core was collected within RC1, corresponding to a gas flare detected by acoustics (32° 9.76′ N, 34° 10.12′). In 2016, we collected an additional core within the pockmark in the western flank of Levant Channel (LC; AG16-15BC1, 32°09.88' N, 34°06.88' E, water depth 1261 m). In 2023, we collected two additional box corer samples near the Palmahim Disturbance brine pool site (BP; Pal1 and Pal2, 32° 13.33' N, 34° 10.72' E and 32° 13.32' N, 34° 10.71' E, ~10 and 20 m away from the brine pool, respectively, water depth ~1135 m). The sediments in each of the box corers were subsampled using ~40 cm-long push corers (60 mm diameter) and sectioned onboard at 1 cm resolution. We also collected distinct samples from the same section, based on the black coloration of the burrow walls and bright coloration of the ambient sediments. The sections for DNA extractions were frozen and kept at -20 °C until further processing. Additional push cores were collected from the same box-core samples, kept intact, and stored at 4 °C for X-ray computerized tomography (CT) scanning.

Table 1: Box core samples used in this study. RC1 is a ridge-crest pockmark system, LC is Levant Channel, BP is a brine pool area (Figure 1). The following analyses were performed: CT scan (1), nutrient profiles (2), flow cytometry (3), activity measurements (4), amplicon sequencing of the 16S rRNA gene (5), metagenomics (6) and dinitrogen fixation (7).

| Sample | Year | Depth (m) | Latitude | Longitude | Location | Burrowing | Analyses |
|--------|------|-----------|----------|-----------|----------|-----------|----------|
| AG16-17BC1 | 2016 | 1034 | 32°09.62′ | 34°10.02′ | RC1 | High | 1,2,3,4,5,7 |
| AG16-15BC1 | 2016 | 1242 | 32°09.89′ | 34° 06.92′ | LC | Low | 1,2,3,4,5,7 |
| Box1 | 2020 | 1035 | 32° 09.60′ | 34°10.00′ | RC1 | High | 4,5 |
| Box2 | 2020 | 1035 | 32° 09.60′ | 34°10.00′ | RC1 | High | 4,5 |
| Box3 | 2020 | 1035 | 32° 09.60′ | 34°10.00′ | RC1 | Low | 4 |
| Flare1 | 2020 | 1035 | 32° 09.76′ | 34° 10.12′ | RC1 | Low | 4 |
| Pal1 | 2022 | 1135 | 32° 13.33′ | 34° 10.72′ | BP | High | 4 |
| Pal2 | 2022 | 1135 | 32° 13.32′ | 34° 10.71′ | BP | Low | 4 |



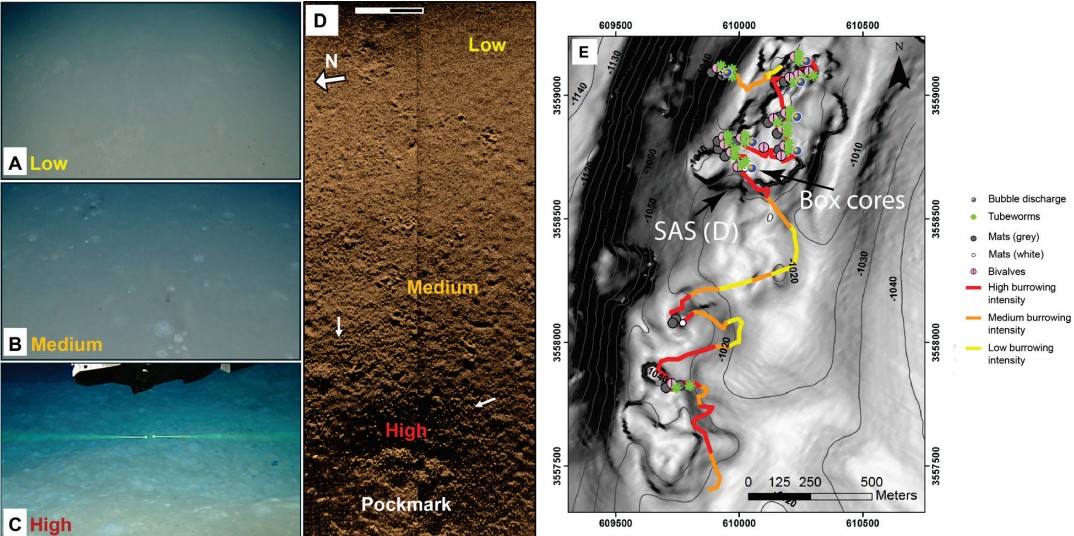


Figure 2. Classification and mapping of burrowing intensities and their correlation with indicators of active seepage. ROV images of low (A), medium (B), and high (C) burrowing intensities are shown (similar scales, with the green laser beams in c. being 7.5 cm apart). D) Synthetic aperture sonar (SAS) backscatter image of the gradient in burrowing intensity from low (top-right) to high (bottom) inside a small pockmark at the edge of the RC1 pockmark system. The burrowing mounds

are observed as a decimeter-scale doted dark-bright pattern (white arrows show examples). E) Map of classified burrowing intensities and active seepage indicators along EV Nautilus 2011 ROV survey tracks of a ridge-crest pockmark system RC1, overlaid on the contours of bathymetry (at 5 m spacing) and a grey-scale map of the bathymetric gradient (high gradients are darker). Locations of the SAS swath in panel D and the cored area are marked by black arrows in panel E.

### 2.4 Computerized tomography (CT) scans

We performed computer tomographic (CT) scanning using a Siemens Somatom Definition AS62 installed at the Institute of Forensic Medicine (University of Bern, Switzerland). Whole core sections were scanned at a spatial resolution of 300 µm, using energies of 120 kV and effective power of 35 mA. Image analyses and processing (combined single thresholding and watershed segmentation, labeling and quantification) were done using Avizo 9.4 (FEI – Thermo Scientific). Before segmentation and labeling, images were filtered using a non-local mean filter. Bioturbation features have been segmented

and quantified for each slice across the full scanned volume. Results are presented as volume % vs. depth (each 400 µm from top to bottom). For the overall core section, bioturbation has been described according to the Bioturbation Index (BI) (Reineck, 1963) and assessed qualitatively (taking a descriptive approach) for approximately every 10 cm interval of boxcore. This method assesses bioturbation on a scale from 0-6, where 0 indicates 0% bioturbation, 1 corresponds to 1-4 % bioturbated sediment, 2 between 5 to 30 % bioturbation, 3 between 31 to 60 % bioturbation, 4 between 61 to 90 %

bioturbation, 5 between 91 to 99 % bioturbation and 6 reflects 100% bioturbation. The shape and size of the burrows were described according to the length and thickness/diameter of the burrowing and their orientation (Jumars et al., 2007).



### 2.5 DNA extraction and sequencing

DNA was extracted from the sediment (~500 mg wet weight) samples with the legacy DNeasy PowerSoil kit (Qiagen, USA), or the FastDNA Spin Kit for Soil (MPBio, USA), using the FastPrep-24™ Classic (MP Biomedicals, USA) bead-

beating to disrupt the cells (2 cycles at 5.5 m sec⁻¹, with a 5 min interval). The V4 region (~ 300 bp) of the 16S rRNA gene was amplified from the DNA (~50 ng) using the 515Fc/806Rc primers (5'- GTGYCAGCMGCCGCGGTAA, 5'-GGACTACNVGGGTWTCTAAT) amended with either CS1/CS2 tags (2020 samples) or SP1/SP2 tags (2023 samples) (Apprill et al., 2015; Parada et al., 2016). PCR conditions were as follows: initial denaturation at 94 °C for 45 s, 30 cycles of denaturation (94 °C for 15 sec), annealing (15 cycles at 50 °C and 15 cycles at 60 °C for 20 sec) and extension (72 °C

for 30 s) (Sisma-Ventura et al., 2022). Library preparation from the PCR products and sequencing of 2x250 bp Illumina MiSeq reads was performed by HyLabs (Israel) in 2020 and Syntezza Bioscience (Israel) in 2021. Five metagenomic libraries were constructed and sequenced to a depth of ~70 million 2x150 Illumina reads at Hylabs (Israel) (2 libraries from the burrows: 9-10 cm section in BOX1, 8-9 cm section in BOX 2; 3 control libraries from BOX1: 1, 2 and 5, 25 and 29 cm below sediment surface).

### 2.6 Bioinformatics

Demultiplexed paired-end reads were processed in QIIME2 V2021.4 environment (Bolyen et al., 2019). Primers and sequences that didn't contain primers were removed with cutadapt (Martin, 2011), as implemented in QIIME2. Reads were truncated based on quality plots, checked for chimeras, merged and grouped into amplicon sequence variants (ASVs) with DADA2 (Callahan et al., 2016), as implemented in QIIME2. The amplicons were classified with scikit-learn classifier that

was trained on Silva database V138 (16S rRNA, Glöckner et al., 2017). Mitochondrial and chloroplast sequences were removed from the 16S rRNA amplicon dataset. Downstream analyses were performed in R V3.6.3 (R Core Team, 2023), using packages phyloseq (McMurdie and Holmes, 2013) and ampvis2 (Andersen et al., 2018).

Metagenomic libraries were processed using ATLAS V2.3 (Kieser et al., 2020), with SPAdes V3.14 de-novo assembler (Prjibelski et al., 2020) with --meta -k 21,33,66,99 flags. We used metaBAT2 (Kang et al., 2015) and maxbin2 (Wu et al.,

2015) as binners, DAS Tool (Sieber et al., 2018) as a final binner for metagenome-assebled genome (MAG) curation. MAGs were dereplicated with dREP (Olm et al., 2017), using a 0.975 identity cutoff. We used CheckM2 to assess the completeness of MAGs (Chklovski et al., 2023), assigned taxonomy with GTDB-Tk V1.5 and GTDB R202 taxonomy (Chaumeil et al., 2020) and annotated them with the SEED and the rapid annotation of microbial genomes using Subsystems Technology (RAST) (Overbeek et al., 2014). Identification of key functions was based on the hidden Markov model

(HMM) profiles within METABOLIC (Zhou et al., 2022). Metagenomic reads and MAGs were deposited to the NCBI under project number PRJNA1072319.

### 2.7 Prokaryote abundance

Total prokaryotes abundance (bacteria and archaea) was quantified from both the upper sediment (top 18-25 cm, every ~2 cm) and above the seabed (50 m, every 5-20 m) using flow cytometry. Sediment samples (~3 gr) were collected into sterile

15 ml plastic tubes containing 6 ml of pre-filtered seawater (0.2 μm) from the same station, fixed with a flow cytometry



grade glutaraldehyde solution (1% final concentration, Sigma, G7651) and kept in 4 °C until analyses within ~2 weeks. At the lab, the chelating agent sodium pyrophosphate (final concentration 0.01 M) and the detergent Tween 20 (final concentration 0.5%) were added. The samples were then vortexed at 700 rpm, placed in an ultrasonic water bath (Symphony, VWR) for 1 min to disperse the grain-attached prokaryotic cells into the liquid seawater phase, and left

overnight at 4 °C. At the lab, subsamples were stained for 10 min in the dark with 1 µl SYBR green (Applied Biosystems cat #S32717) per 100 µl of the sample. The samples were analyzed by an Attune® acoustic focusing flow cytometer at a flow rate of 25 µL min$^{-1}$. Taxonomic discrimination was based on cell side scatter, forward scatter, and green fluorescence. Cell abundance was normalized to the sediment's dry weight.

### 2.8 Heterotrophic activity

Samples of ~3 gr were collected across the upper sediment (18-25 cm, see subsection 2.5), re-suspended in 3.5 ml of pre-filtered seawater/porewater (0.2 µm) and spiked with 500 nmol L$^{-1}$ (final concentration) of [4,5-$^3$H]-leucine (Perkin Elmer USA, specific activity 160 Ci mmol$^{-1}$). Porewater from 5-25 cm was first degassed from O$_2$ by bubbling into them N$_2$ for ~10 min, thus keeping them anoxic. The samples were incubated in the dark for 4-5 h under in situ temperatures. At the end of the incubation, 100 µl of 100% trichloroacetic acid solution was added to stop any microbial assimilation of leucine,

followed by sonication (Symphony, VWR) for 10 minutes to remove bacterial biomass from the sediment grains (Frank et al., 2017). The bacterial biomass extract from the liquid phase and divided into three 1 ml aliquots. For the seawater samples, the collected material was added with 100 nmol L$^{-1}$ of [4,5-$^3$H]-leucine (same working solution as above, final concentration), incubated for 4-5 h in the dark, and 100 µl of 100% trichloroacetic acid solution was added to stop the incubation. The sediment's liquid extracts were then processed using micro-centrifugation (Smith et al., 1992).

Disintegration per minute (DPM) from each sample was read using a Packard Tri carb 2100 liquid scintillation counter. A conversion factor of 1.5 kg C mol$^{-1}$ with an isotope dilution factor of 2.0 was used to calculate the carbon assimilation rate (Simon and Azam, 1989). Blanks included sediments added with [4,5-$^3$H]-leucine and immediately with trichloroacetic acid, incubated and processed under the same conditions as described above.

### 2.9 Nitrogen fixation rates

We evaluated N$_2$ fixation rates in the top sediment (upper 3 cm) and overlying seawater ~1 m above the seabed, in push cores AG16-15BC1 and AG16-15BC1 sampled in 2016.  The sediment samples (~150 ml) were placed in 250 ml gas-tight bottles filled with porewater from the same sediment layer (~100 ml) and $^{15}$N$_2$ enriched seawater (10% of the water volume), as previously described (Mohr et al., 2010). In short, $^{15}$N$_2$ gas (99%, Cambridge Isotopes) was injected at a 1:100 ratio (*v:v*) to filtered (0.2 µm, PALL) and degassed (MiniModule G543) seawater (FSW) or porewater (FPW) and shaken vigorously

for ~12 h to completely dissolve the $^{15}$N$_2$ gas. The sample bottles were incubated in the dark under an ambient temperature of ~15 °C for 168-172 h. After incubation, the bottles were vortexed for 30 min at 700 rpm, placed in an ultrasonic water bath (Symphony, VWR) for an additional 10 min, left on the bench for ~1 h, and filtered using pre-combusted GF/F filters, following 100 µm mesh pre-filtration to remove large aggregates. For the overlying seawater measurements, 4.5 L were placed in darkened Nalgene bottles with $^{15}$N$_2$ enriched seawater (10% of the water volume). The bottles were incubated for

72 h under ambient temperature and the content was filtered onto pre-combusted GF/F filters. Triplicate bottles without



$^{15}$N enrichment were used for determining the natural abundance of the particulate matter at each station. The filters were kept at -20 °C in sterile petri plates until dried at 60 °C overnight and analyzed on a CE Instruments NC2500 elemental analyzer interfaced with a Thermo-Finningan Delta Plus XP isotope ratio mass spectrometer (IRMS). For isotope ratio mass spectrometry, a standard curve to determine N mass was generated for each sample run. Based on natural abundance,

N mass on the filters, incubation times, and precision of the mass spectrometer, our calculated detection limit for $^{15}$N uptake was 0.01 nmol N L$^{-1}$ d$^{-1}$. N$_2$ fixation rates were calculated based on equations 2 and 3 after Mulholland et al., (2006) using N solubility factors of Weiss (1970).

Equation 2
$$N2\ fixation = \frac{[\text{Atom\% (final)} - \text{Atom\% (t0)}]}{[\text{Atom\% N2} - \text{Atom\% (t0)}]} x \frac{\text{PN(final)}}{\Delta t}$$

Equation 3
$$\text{Atom\%} = \frac{15N}{(15N + 14N)} x100$$

Where the Atom%$_{(final)}$ and Atom%$_{(t0)}$ represent the fractional $^{15}$N enrichment of the particulate nitrogen (PN) pool after the incubation and of the ambient seawater, respectively. Atom% N$_2$ is the fractional $^{15}$N enrichment of the N$_2$ source pool.

PN$_{(final)}$ is the concentration of the particulate nitrogen at the end of the incubation, and $\Delta t$ is the incubation length (72 h for the seawater and 168-172 h for the sediment incubations).

## 3. Results and Discussion

### 3.1 Burrowed sediments are concentrated around Palmahim Disturbance seepage hotspots

While burrow mounds are rare in the generally smooth seafloor in the bathyal SEMS, these seabed features were particularly

abundant at the toe region of Palmahim Disturbance. The seabed burrow extensions appeared in two main forms: i) decimeter-scale circular sediment mounds with mostly bright color, often similar to that of the surrounding seafloor and ii) several millimeters to decimeter-scale holes, commonly surrounded by scrape marks (Figure 2). The borrow density varied from only a few sporadic borrows (low intensity) to a complete convolution of the seafloor within an ROV field of view (high intensity; Figure 2). In the regions of intense burrowing, we observed rugged seafloor with localized depressions and

mounds (bright color, often dark heaps).

The video analysis and mapping of active seepage markers along the ROV dive tracks, as well as the SAS surveying, revealed that bioturbation intensities increased towards the inner part of a pockmark (Figure 2E). The high burrowing intensity is often associated with the presence of seepage indicators, including microbial mats, bubbles, or chemosynthetic fauna. Within the large pockmark RC1, nearly the entire soft sediment seafloor was intensely bioturbated, except in the

vicinity of rock outcrop. The domes of intensely burrowed sediments reached over a meter in diameter and sometimes overly neighboring ones. Our observations suggest that burrowing is markedly enhanced by nearby seepage, whereas the burrowed areas largely exceed the vicinity of visible seepage hotspots, often beyond 100 m distances.





### 3.2 *Calliax lobata* ghost shrimps inhabit near-seep large burrows

Box core sediment samples within RC1 area (AG16-17BC1, BOX 1 and BOX2) were intensely burrowed (Figure 3). The sediment surface within these samples appeared fluffy with decimeter-scale mounding and multiple holes and scratches (Figure 3A). We observed sporadic black coloration at various horizons, indicative of large, ~0.5 cm wide burrows (Figure 3B-D). Random sieving of leftover sediments in the respective box-corer with a 2 mm mesh revealed that macrofauna was usually scarce, except for 6 live specimens *Calliax lobata* ghost shrimp, measuring ~5 cm long and ~1 cm wide (Figure 3E). *C. lobata* claws consistently dominated the >2 mm sieved fraction, varying in size up to ~1 cm long and ~5 mm wide (Figure 3F), following previous observations in the same area (Beccari et al., 2020). We, therefore, suggest that *C. lobata* has a key role in near-seep sediment burrowing.

CT scans of AG16-17BC1 and AG16-15BC1 cores showed considerable burrowing (Figure 4). The burrows were imaged as lower density to empty, sub-vertical to sub-horizontal, linear and helicoidal, and generally cylindrical tubes that vary in their diameter between ~1 to ~10 mm. Larger burrows were usually surrounded by bright halos or rings of relatively dense material. Several burrows had chamber-like regions (Figure 4c). Pervasive tone variations across the CT scans suggest the impact of multiple generations of burrowing that convolute a major part of the cored sections, with more pronounced tone-contrasts imaging the most recent generation of burrowing.

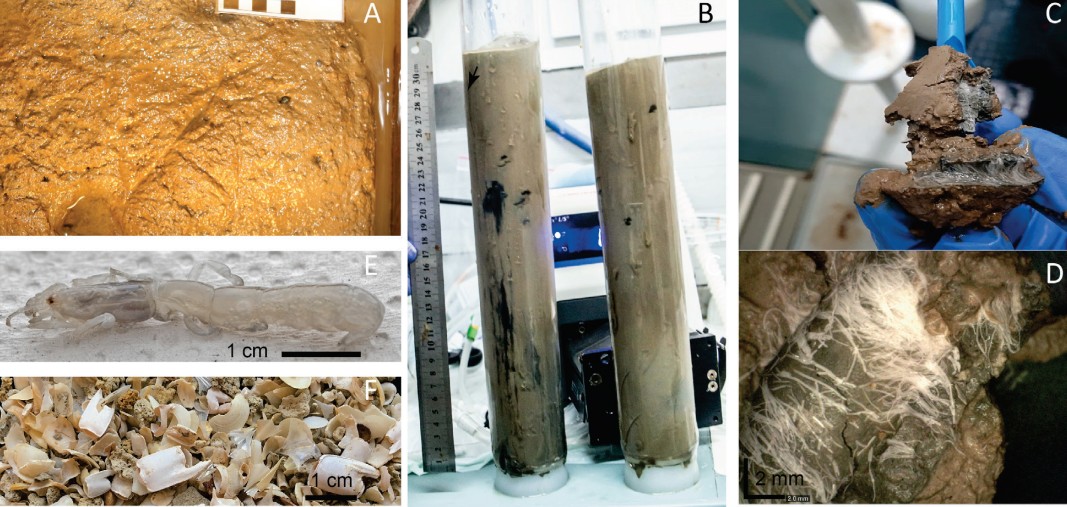

Figure 3. Burrows of the ghost shrimp *Calliax lobata* near active seepage sites within Palamhim Disturbance in the Eastern Mediterranean Sea. A) The seafloor, as exposed in the AC16-17BC1 box-core (RC1 area, 2016), shows burrowing mounds and openings, as well as scratches on the sediment surface. B) Image of BOX1 and 2 cores (RC1 area, 2020), showing dark coloration of burrows, unusual for the ambient sediments. C and D) Zooming into the burrows of BOX1 and 2 cores showed the presence of large filamentous microbes. E) Live *C. lobata* specimen found in AC16-17BC1. F) A multitude of *C. lobata* claws recovered through sporadic sieving (2 mm mesh) of AC16-17BC1.



Segmentation of the CT images captures these more pronounced contrasts, marking the current burrowing activity. Core AG16-17BC1 (47 cm recovery) was composed of homogeneous oxidized clayey sediments with a high amount of shell

fragments and pteropod fragments. In this core, the burrow segmentation had an average value of 0.9 %, with a maximum of 7.0 % of the total sediment volume. We noted the highest burrowing at 5 cmbsf (~ 4%) and between 20-22 cmbsf (~ 7%). Between 0 and 15 cmbsf, the sediments had an overall sparse burrowing (BI = 1) with a few discrete traces. Discrete traces were composed of elongated sub-vertical burrows having a length of approximately 3 cm and a diameter of ~1-5 mm, as well as helicoidal burrows similar in size. One large vertical burrow being 12 cm in length and with a diameter of

1 cm is present from the surface to 15 cmbsf. From 15 cmbsf to 25 cmbsf, the overall BI was again close to 1 with few discrete traces composed of very small sub-vertical burrows being 1 cm in length and 1 mm in diameter, as well as two large-sized burrows: 15 cm in length (1 cm diameter) and 6 cm in length (4 cm thickness). From 25 to 41 cmbsf, no clear bioturbation features were quantified (BI = 0).

Core AG16-15BC1 was composed of oxidized clayey sediment with minor horizontal layering. The surface was highly

bioturbated showing a wavy and slightly disturbed surface with small mounds, as well as a high amount of pteropod shells. The average volume % burrowing was 0.7 % with peak burrowing at 4 cmbsf and 10 cmbsf (up to 4%). The box-core sediments (0-40 cmbsf) present overall sparse burrowing (BI = 1). From 0-20 cmbsf, a few discrete traces (BI = 1) were composed of elongated sub-vertical burrows of 5 cm in length and 0.5 cm in diameter and horizontally elongated burrows of 6 cm in length and 0.5 cm in diameter. Also, small curved burrows were present (1cm in length and 1-2mm thick). From

20-35 cmbsf, few discrete traces (BI = 0-1) were composed of vertically elongated burrows of 5 cm length and 2-3 mm diameter, as well as small curved burrows of 2 cm length and 1-2 mm thickness with sub-vertical orientation. The amount of bioturbation decreased towards 35 cmbsf. From 35 – 40 cmbsf, no bioturbation (BI=0) was visible.

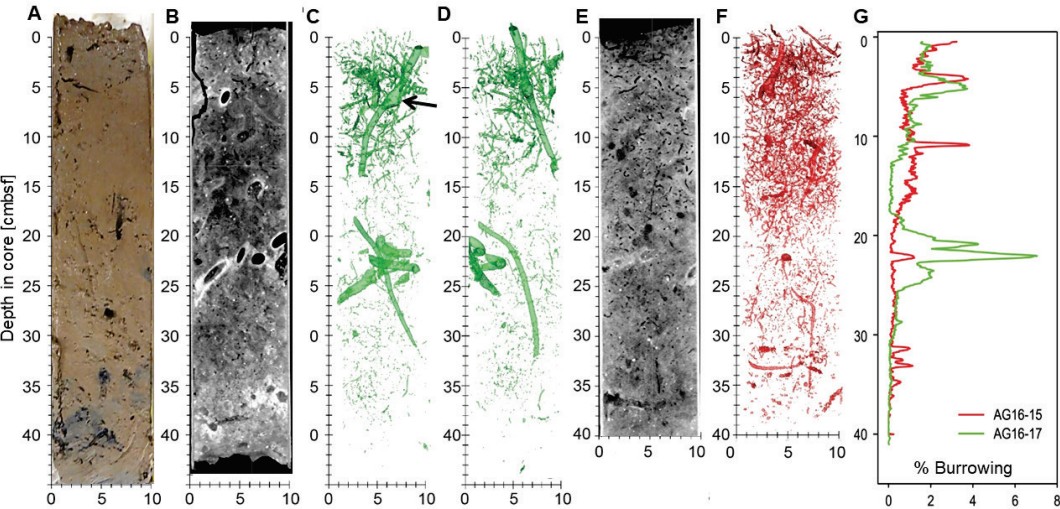



Figure 4. Burrowing in AG16-15BC1 and AG16-17BC1, as detected by computerized tomography (CT) scans. A) A vertical section through a sub-core of box-core AC16-17BC1 cut and imaged immediately after sampling. B) A vertical section through the 3D CT scan of another sub-core of AC16-17BC1 (standard medical CT display where higher relative X-ray intensities are darker and low are brighter, depicting lower and higher material densities, respectively). C and D) Two perpendicular 3D displays of segmentation of the relatively vacant (high intensity) burrows, imaged in the CT scan of the same core (in B). E) A vertical section through the 3D CT scan of a sub-core of AC16-15BC1 (displayed as in B). F) A 3D display of AC16-15BC1 segmentation of the vacant (high intensity) burrows. G) Graphs comparing the volumetric % of relatively vacant burrows in CT scanned sub-cores of AC16-17BC1 (green) and AC16-15BC1 (red), as estimated based on the respective segmentation results (in C-D and F).

Based on the burrow morphology, three distinct burrow traces/ichnofabrics were distinguished: i) large (≥5 mm wide) vertical to sub-horizontal burrows (*Thalassinoides*); ii) smaller (2-5 mm wide) vertical and sub-horizontal burrows forming a dense network (*Ophiomorpha*), iii) small (≤2 mm wide) curved and helicoidal burrows (*Gyrolithes*) (Seilacher, 2007). The presence of significant sub-horizontal segments and chambers in the *Thalassinoides* burrows suggests they were created by arthropods. First, it is unlikely that sub-horizontal burrows would be created by a bivalve, which would usually produce vertical to oblique burrows (Trueman et al., 1966; Knaust, 2015). Second, arthropods, and in particular decapods, excavate chambers that allow them to turn their direction within their burrows (Stamhuis et al., 1997; Hembree, 2019; Coelho et al., 2000). In turn, the sparse spacing of the burrows also suggests a decapod rather than a wormlike animal, which often produces tightly packed burrows (Seilacher, 2007). We suggest that *C. lobata* is the primary inhabitant of the two larger *Thalassinoides* and *Ophiomorpha* burrow morphologies, prominent in AG16-17BC1, based on the marked abundance of these animals and matches in size. It is impossible to determine at this stage the provenance of the smaller *Gyrolithes*-type burrows, possibly worms-like animals.

### 3.3 Burrowing alters geochemical processes at seep chemotones

Our data indicates that intense near-seep burrowing can affect geochemical processes via the exchange of solutes across the sediment and the overlying water. The measurements of nutrient concentrations indicate that $NO_3^-$ depth profiles in cores AG16-15BC1 and AG16-17BC1 show different penetration depths and standing stocks, correlated to the presence of burrows (**Figure 5A,** we assumed negligible nitrite concentrations). In core AG16-15BC1, we observed an increasing trend of $NO_3^-$ concentrations from ~35 cm upwards. In comparison, core AG16-17BC1 showed significant enrichment above 20 cm depth. Assuming steady-state conditions, the amount of $NO_3^-$ in the upper 35 cm of core 15 doubled that of core 17 (~260 and ~105 nmol $NO_3^-$ depth cm$^{-2}$, respectively). Nitrification driven by ammonia-oxidizing archaea is the key process in background sediments of the abyssal SEMS (Rubin-Blum et al., 2022). However, in the seepage area, this process may be regulated by burrowing, altering the diffusive access of ammonium from the ambient sediment and supplies of oxygen via irrigation by the burrow inhabitant (Kristensen and Kostka, 2005). Thus, macrofaunal burrowing near the sediment surface may aid nitrification (Mayer et al., 1995; Glud et al., 2009). The gradual upward change in the $NO_3^-/PO_4^{3-}$ ratios (**Figure 5B**) further supports the enhancement of nitrification. Nonetheless, *C. lobata* burrowing may enhance denitrification (removal of nitrate), which was suggested to be enhanced and correlated with the presence of burrowing macrofauna (Nielsen et al., 2004). In core AG16-17, denitrification is likely prominent, given the rapid depletion of nitrate (**Figure 5A**). Our results are thus in line with the previous observations of infauna and $NO_3^-$ depth distribution at 1450 m



water depth using precise microprofiling, which suggested that burrow irrigation may stimulate both the $NO_3^-$ production and consumption (Glud et al., 2009).

Previous observations suggest that nitrifying bacteria have a positive correlation to Fe(III) and a negative correlation to dissolved sulfide, which inhibits nitrification (Kristensen and Kostka, 2005). Experimental studies supported field data,
confirming that increased amounts of Fe(III) react with sulfide, intensifying the effect of nitrification-denitrification processes. In both cores, the depth distribution of porewater $SO_4^-$ concentrations show values slightly lower than the ambient seawater (31 mM; Herut et al., 2022), suggesting the occurrence of sulfate reduction (**Figure 5C**). Nonetheless, the sulfate levels in core AG16-17BC1 are consistently lower than in AG16-15BC1, indicating that sulfate reduction in this core, where *C. lobata* burrows were found, was enhanced. This is in line with previous studies, showing that sedimentary infaunal
burrows and irrigation impact the depth distribution of early diagenetic dissolved reactants/electron acceptors (e.g. $O_2$, $NO_3^-$, and $SO_4^{2-}$) and products (e.g. $CO_2$, $NH_4^+$, and $H_2S$) in porewaters (Aller, 2001; Kristensen and Kostka, 2005; Meile and Cappellen, 2003). The metabolic potential of burrow-associated microbial communities for sulfate reduction and nitrogen cycling in the burrows is further displayed by metagenomics.

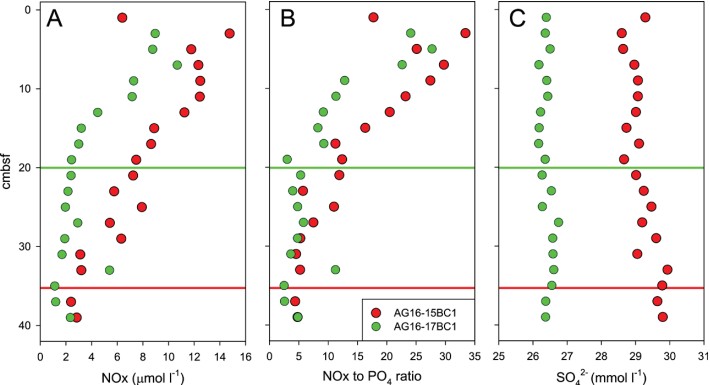

**Figure 5**: Pore water characteristics in cores AG16-15BC1 and AG16-17BC1: A) Nitrate+nitrite (NOx) concentration. B) NOx to phosphate ratio, C) Sulfate concentration. Red and green lines indicate the depths at which NOx concentrations appear to stabilize.

**3.4 Burrowing may alter microbial abundance and activity at seep chemotones**

Following the previous observations of elevated microbial activity in seep chemotone sediments (Sisma-Ventura et al.,
2022), our measurements in core AG16-17BC1 suggest a weak, but feasible link between burrowing and microbial activity in these sediments (**Figure 6**). While the highest abundance and activity values were found in the upper 5 cm, we observed a substantial variation in activity below 5 cm in this core (**Figure 6A**). In sediments deeper than 5 cm, we observed a trend linking burrow intensity and the rate of leucine uptake (**Figure 6B, C**). We note that activity was not estimated in burrows, but in random subsamples from the respective sections, thus may reflect only partial burrow effects.




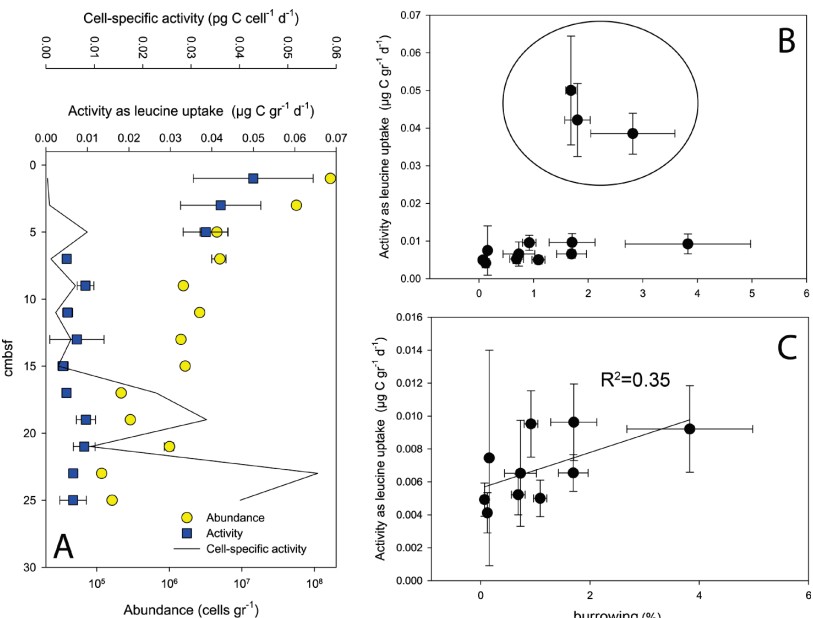

**Figure 6**: Abundance and activity of microbes in core AG16-17BC1: A) Abundance and activity profiles. B) Activity versus burrowing. C) Activity versus burrowing – upper 5 cm removed.

### 3.5 Seep chemotones host unique microbial communities that are altered by burrowing

Amplicon sequencing of the 16S rRNA gene highlighted the unique microbial communities that existed at seep chemotones (**Figure 7, Supplementary Figure S1**). These communities markedly differed from those found in seepage hotspots in the same area (Rubin-Blum et al., 2014b), and those of background sediments in the deep SEMS (Rubin-Blum et al., 2022). We identified different levels of seepage-related perturbations in the sediments. Flare1 and Pal1 samples represent the poorly disturbed, typical background sediments in the SEMS. The most altered sediments were in Box1, collected within

the vicinity of authigenic carbonate in 2020 (Sisma-Ventura et al., 2022) and Pal2, collected 40 meters away from the brine pools in 2022. Ammonia oxidizers such as Nitrosopumilaceae and Nitrospiraceae, as well as other microbes prominent in background SEMS sediments, such as Thermoplasmata and Binatota lineages (Rubin-Blum et al., 2022), were depleted in these profiles, whereas the putative methane oxidizers, sulfate reducers and chemoautotrophs were found (**Figure 7, Supplementary Figure S1**). In particular, the anaerobic methane oxidizers ANME-1 were found at depths below 25 and 5

cm, in Box1 and Pal2, respectively. The aerobic Methylomonadaceae methane oxidizers were prominent in the upper sections of these cores. The read abundance of Desulfosarcinae sulfate reducers increased with distance from the sediment surface, and the putative chemoautotrophs Thiotrichales were often found in these samples. These observations provide evidence for chemosynthetic activity in these chemotone sediments, following previous observations of enhanced microbial activity (Sisma-Ventura et al., 2022).

These results are in line with visual inspection and microscopic observations that revealed: i) black coloration of burrow walls, potentially indicative of precipitation of minerals following sulfate reduction and ii) large, white filaments typical of




giant sulfur bacteria inhabiting the burrow inner perimeter (Ionescu and Bizic, 2019) (**Figure 3 C-D**). According to amplicon sequencing, the key families that were prominent specifically in the burrow subsamples included Desulfobulbaceae, known mainly as sulfur and/or cable bacteria (Trojan et al., 2016), Sulfurovaceae autotrophic sulfur

oxidizers, often found in association with marine invertebrates (Bai et al., 2021; Hui et al., 2022) and Flavobacteriaceae versatile degraders of macromolecules, especially carbohydrates (Gavriilidou et al., 2020; Chen et al., 2021). Similar key taxa appear to be associated with bioturbation in other habitats, such as the intertidal flats (Fang et al., 2023). These findings lead to the hypothesis that unique metabolic handoffs take place within the burrows, altering carbon, nitrogen and sulfur cycling.

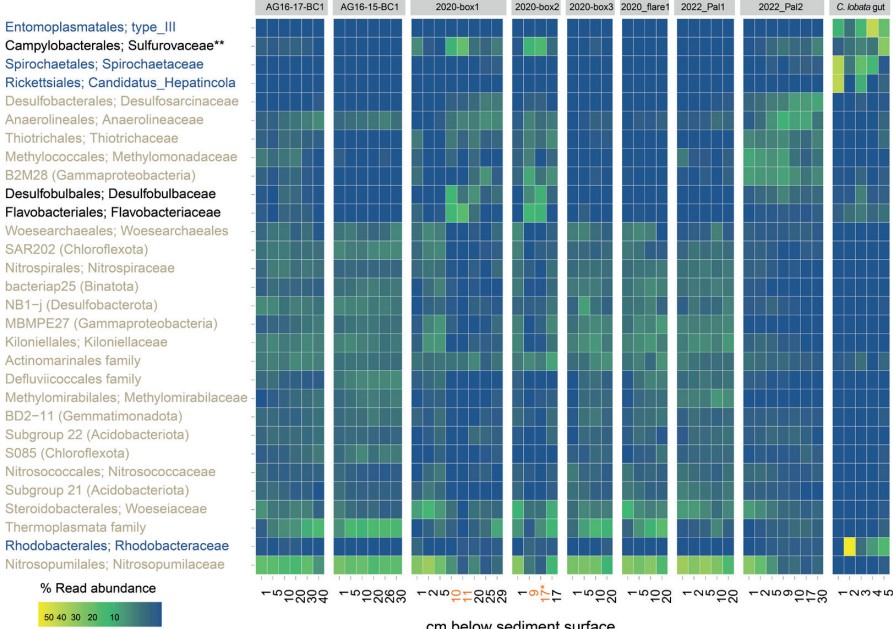


**Figure 7**: Read abundance of the top 30 most prevalent prokaryote taxa at the family level, in Palmahim sediments (see Table 1) and the hindgut of the ghost shrimp *Calliax lobata* (black: enriched in burrows, blue: enriched in *C. lobata*, brown: enriched in sediments). Sediment depths where burrow microbes were collected are highlighted in orange. * Both samples are from the 16-17 cm section, yet the first one is taken from the burrow and the second from background sediments. **

abundant in burrows and associated with *C. lobata* hindgut.

### 3.6 Metabolic handoffs at burrow walls may fuel dark productivity

Based on two DNA samples extracted specifically from burrows in cores Box1 and Box2, we curated 58 high-quality metagenome-assembled genomes spanning 15 phyla (57 bacteria, 1 archaeon, median quality 92%, median contamination 2%, **Figure 8**). We observed considerable genotype diversity within Bacteroidota, Desulfobacterota and Proteobacteria

phyla (mostly Gammaproteobacteria in the latter). 51 MAGs were found in both samples, although the read abundance patterns often differed. In particular, we found the Thiohalobacteraceae (MAGs 20 and 113, reclassified as UBA9214 by GTDB) and *Thiomargarita* (MAG 115) autotrophic sulfur oxidizers in Box2, in which the giant bacteria were observed, whereas Thiohalomonadaceae (MAG 83), Sedimenticolaceae (MAGs 22 and 46), and Sulfurimonadaceae



(Campylobacterota, MAGs 2 and 80) appear to be the key autotrophs in Box1. Flavobacteriaceae (mainly CG1-02-35-72 lineage, MAGs 10 and 69) were abundant in both samples, but most dominant in Box1 (14.4% read abundance). Desulfobacterota comprised Desulfobacterales (MAGs 55 and 75), and mainly Desulfobulbales lineages BM004 and BM506, associated with the dissimilatory reduction of perchlorate (Barnum et al., 2018), as well as *Desulfocapsa*, known to specialize in disproportionating inorganic sulfur compounds (Finster et al., 2013; Ward et al., 2021). BM004 MAG 88 was the dominant one in Box2 (4.1% read abundance) and *Desulfobacula* (Desulfobacterales) MAG 55 In Box1 (4.7% read abundance). Themodesulfovibrionia UBA6902 (Nitrospirota) MAG 11 was also found in Box2 (2.2% read abundance). In Box1, Anarerolineales UBA11858 MAG 99 was prominent (4.5% read abundance). Gemmatimonadetes (MAGs 9 and 89, ~5% read abundance each) were abundant in Box2. We hypothesized that these dominant lineages can perform an array of metabolic functions, using a range of substrates and electron donors/acceptors, markedly altering the functionality of ecotones compared to the unaltered bathyal sediments in the SEMS.

In both samples, the chemotrophs that can fix inorganic carbon comprised most gammaproteobacterial lineages and Campylobacterota. Most of these organisms may fix carbon via the Calvin–Benson–Bassham (CBB) cycle, given the presence of *cbbLS* and/or *cbbM* genes encoding the form I and II ribulose-1,5-bisphosphate carboxylase/oxygenase (RubisCO), respectively (**Figure 8**). Sulfurimonadaceae MAGs encoded the form I ATP citrate lyase, needed for carbon fixation via the reverse tricarboxylic acid cycle (rTCA). In agreement with previous studies (Flood et al., 2016), *Thiomargarita* encoded not only the form II RubisCo but also the form II ATP citrate lyase, indicating the potential to fix carbon via more than one pathway (Rubin-Blum et al., 2019). Partial or complete Wood-Ljungdahl (WL) pathway was widespread in Chloroflexota, Desulfobacterota and Nitrospirota, suggesting the potential for carbon fixation using energy from dihydrogen oxidation and sulfur reduction (Drake, 1994). Alternatively, some of these organisms may use this pathway for energy conservation during acetoclastic growth (Alves et al., 2020; Santos Correa et al., 2022; Fang et al., 2022).

Either oxidation of reduced sulfur, or sulfur/thiosulfate disproportion may be the key drivers of chemoautotrophy in the burrows, especially by Proteobacteria and Campylobacterota. This is indicated by the widespread occurrence of genes encoding proteins needed for dissimilatory sulfur cycling, such as Sqr, DsrAB, AprAB, components of the Sox system, as well as the PhsA, a key protein in thiosulfate disproportion (Finster, 2008; Anantharaman et al., 2018) (**Figure 8**). While oxygen is the likely electron acceptor for chemosynthesis, denitrification appears to play a substantial role in energy metabolism. This is indicated by the frequent occurrence of genes needed for dissimilatory nitrogen cycling, e.g., those encoding Nap and Nar (nitrate reduction), NirK, NirS and NirBD (nitrite reduction, as well as NrfA for dissimilatory nitrite reduction to ammonia, DNRA), NorBC (nitric oxide reduction) and NosZ (nitrous oxide reduction to dinitrogen) (**Figure 8**). Most gammaproteobacterial MAGs encoded all or part of these genes, indicating that denitrification could be completed either by an individual lineage or synergistically.




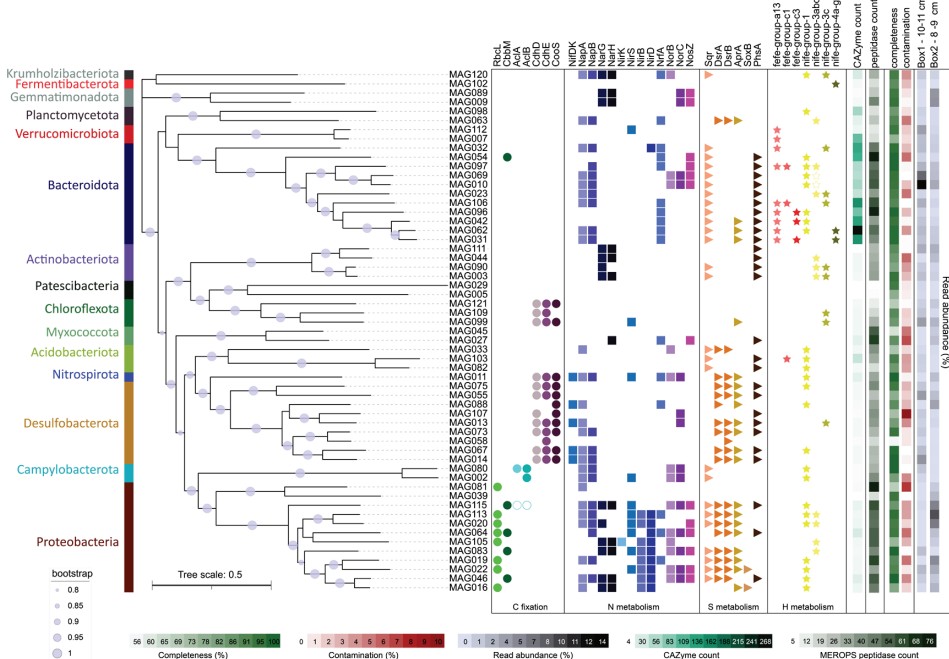

**Figure 8**: Phylogeny, taxonomy and key functions of bacteria, based on the reconstruction of metagenome-assembled genomes (MAGs) from *Calliax lobata* burrows. FastTree phylogeny is based on the GTDB-tk marker gene set. Functions were assigned using METABOLIC Hidden Markov Model (HMM) set. Empty circles and stars indicate the functions discovered through manual annotation of functions, using SEED and NCBI BLAST.

In turn, the sulfur cycling can be driven by fermentation, evolving hydrogen, or other products of this process, such as hydrogen, acetate, butyrate and lactate (Campbell et al., 2023; Ruff et al., 2023). Similar to groundwater communities (Ruff et al., 2023), the key fermenters of macromolecules appear to be the highly abundant Bacteroidota (**Figure 8**). These organisms are ubiquitous degraders of macromolecules such as complex glycans (Martens et al., 2009; Ndeh and Gilbert, 2018; Martens et al., 2008; Fernández-Gómez et al., 2013). They encode carbohydrate-active enzymes (CAZymes) that are often organized in polysaccharide utilization loci (PULs), with typical SusCD transporters (Martens et al., 2009; Reeves et al., 1996; Grondin et al., 2017). We found multiple SusCD in copies in diverse Bacteroidota MAGs, but also discovered a multitude of encoded CAZymes, as well as peptidases, highlighting the potential of these bacteria to use a broad scale of macromolecules.

Bacteroidota may ferment these macromolecules to produce various substrates, most importantly, hydrogen, as suggested by the presence of Fe-Fe hydrogen evolving hydrogenases in Bacteroidales MAGs 31, 42, 62, 96 and 106 (**Figure 8**). The highly abundant Flavobacteriales MAGs 10 and 69 did not encode the Fe-Fe or the canonical type 4 $H_2$ evolving hydrogenases. Instead, they encoded hydrogenase I (*hyaABCDEF*), and, most importantly, an adjacent gene cluster encoding an aberrant putative HoxHYFU NAD-dependant hydrogenase, apparently widespread in Bacteroidota (based on multiple NCBI nr BLAST hits of HoxH protein in this clade, data not shown). These soluble, oxygen-tolerant, bidirectional hydrogenases described in *Ralstonia eutropha* H16 (Burgdorf et al., 2006) and cyanobacteria (Appel et al., 2000; Gutekunst



et al., 2014), can promote hydrogen evolution in bacteria that ferment sugars (Cheng et al., 2019). We also identified *fdhAB* genes encoding the NAD-dependent formate dehydrogenase in these Flavobacteriales MAGs, hinting at the potential coupling of formate oxidation and hydrogen evolution. Other genes involved in fermentation, including those needed for
acetogenesis and acetate conversion to acetyl-CoA (*ack*, *pta* and *acs*), were found in all the Bacteriodota MAGs. Thus, most Bacteriodota that we discovered in burrows, including the dominant CG1-02-35-72 Flavobacteriales lineages, are likely the key fermenters that fuel the cycling of organic matter at the burrow walls.

Fermentation products can provide electron donors Desulfobacterota and Nitrospirota sulfate reducers, which encode the Dsr/Apr systems, highlighting the key role of sulfur cycling in these communities (**Figure 8**). This is in line with the
observation of hydrogenogenic fermentation and hydrogenotrophic sulfate reduction that co-occur in anoxic sandy sediments (Chen et al., 2021). These interactions between Bacteroidota and Desulfobacterota are not limited to marine sediments, but for example, facilitate dihydrogen turnover in the human gut (Thomas et al., 2011; Wolf et al., 2016; Gibson et al., 1993).

Hydrogen can also directly fuel chemosynthesis, for example, by Campylobacterota, which encode uptake NiFe
hydrogenases, and often thrive based on hydrogen in hydrothermal vents (Molari et al., 2023). Some Gammaproteobacteria, including the symbiotic Sedimenticolaceae and Thioglobaceae, also use hydrogen as an energy source in chemosynthetic habitats (Petersen et al., 2011). In line with these observations, we identified uptake NiFe hydrogenases in all the diverse Sedimenticolaceae MAGs (16, 19, 22, 46 and 155), as well as in Thiohalobacteraceae MAGs 20 and 113, hinting that hydrogen is a potential energy source for burrow autotrophs.

Our results suggest the presence of productive burrow communities, that cycle carbon, sulfur and nitrogen, potentially through the turnover of hydrogen or other fermentation products. Examples of similar communities, comprising Bacteroidota degraders of macromolecules, sulfate reducers (Desulfobacterota and others), and chemotrophs (mainly Campylobacteria and Gammaproteobacteria) are common among a wide range of biomes, including burrowed sediments (Fang et al., 2023), epibionts of invertebrates in chemosynthetic habitats (Xu et al., 2022; Bai et al., 2021) and
endosymbiotic communities in *Idas* mussels (Zvi-Kedem et al., 2023). The interactions between these organisms are not limited to the exchange of key metabolites (Zoccarato et al., 2022). One example of synergy in this system is the potential exchange of cobalamin (vitamin B12) among the community members. Our data suggest that the taxonomic diversity of cobalamin producers in the burrow walls is limited, primarily to Desulfobacterota and some thiotrophs (*Thiomargarita* MAG 115 and Thiohalomonadaceae MAG 43), as well as several others (**Figure 8**). On the other hand, the key
Flavobacteriaceae (MAGs 10 and 69) do not produce cobalamin but may take it up, as they encode the outer membrane vitamin B12 receptor BtuB and BtuFCD components of the vitamin B12 ABC transporter, providing this cofactor to B12-dependent enzymes such as methylmalonyl-CoA mutase (EC 5.4.99.2) and class II ribonucleotide reductase (EC 1.17.4.1). The resilience of these communities may depend on redundancy in the production of such shared commodities, for example, cobalamin production by diverse Desulfobacterota genotypes.




**3.7 *Calliax lobata* hosts specific gut microflora but may ingest burrow microbes**

The amplicon sequencing of DNA extracted from hindgut or midgut in several *C. lobata* specimens identified specialized microflora, but also microbes found in the burrow walls (**Figure 7, Supplementary Figure 1**). The four key lineages associated with the gut sections included Entoplasmatales, Spirochaetales, Rickettsiales and Rhodobacterales. For example, we found abundant *Candidadus* Hepatincola (Rickettsiales), a common nutrient-scavenging bacterium in the gut of crustaceans, and isopods in particular, from terrestrial and aquatic environments (Dittmer et al., 2023). We found that some lineages, which were prominent in burrow walls, may occur in *C. lobata* gut. For example, we found trace quantities of *Desulfobulbus* and *Desulfocapsa* in the gut of some *C. lobata* specimens (**Supplementary Figure S1**). Campylobacterota were abundant in the gut and the burrow walls, whereas *Sulfurospirillum* species were associated only with the gut, and *Sulfurovum* were found in both the burrows and the gut, indicating that these lineages occupy distinct niches. Altogether, these findings hint that *C. lobata* may be feeding in the burrows, providing evidence for the 'gardening' hypothesis.

**3.8 Chemotone sediments are potential hotspots of dinitrogen fixation**

Our metagenomic data indicates that some members of burrow communities, in particular Desulfobacterota and Notrospirota, have the genetic potential to fix dinitrogen (**Figure 8**). This is in line with the previous observations of enhanced nitrogen cycling by bioturbation in nearshore sediments (Laverock et al., 2011). Hydrocarbon seeps themselves are hotspots of $N_2$ fixation, where benthic diazotrophs can provide up to 30% of the community anabolic growth requirement for nitrogen (Dekas et al., 2014, 2018). Desulfobacterota are prominent diazotrophs in gas-rich sediments, but anaerobic methane oxidizers (ANME), as well as other lineages also play a role in this process (Orphan et al., 2001; Dekas et al., 2018). Our data hint that nitrogen fixation can also extend to the SEMS chemotones, where inputs of new nitrogen may be crucial to sustaining productivity under oligotrophic conditions.

Although we did not measure nitrogen fixation rates directly in the burrows, the $0.02\pm0.004$ fmol N $gr^{-1}$ $d^{-1}$ estimates in the top 3 cm of the chemotone sediments (AG16-17BC1) were significantly higher than the $0.01\pm0.005$ fmol N $gr^{-1}$ $d^{-1}$ background values (AG16-15BC1, ANOVA followed by a Fisher LSD multiple comparison post hoc test, p=0.04, **Figure 9A**). Given that most putative diazotrophs were found in burrow walls, our results likely underestimate its rates in the sediments. $N_2$ fixation rates were also enhanced in the water column approximately 1 m above the chemotoene sediments ($0.04\pm0.01$ nmol N $L^{-1}$ $d^{-1}$ as opposed to $0.01\pm0.009$ nmol N $L^{-1}$ $d^{-1}$ at the control site, ANOVA followed by a Fisher LSD multiple comparison post hoc test, p=0.02, **Figure 9B**), with different diazotrophic populations found at that habitat than those in the sediments (Sisma-Ventura et al., 2022). Altogether, our results suggest that nitrogen cycling at chemotones is substantially altered, either by diazotrophy or denitrification (see above), and that these processes may be boosted by burrowing.



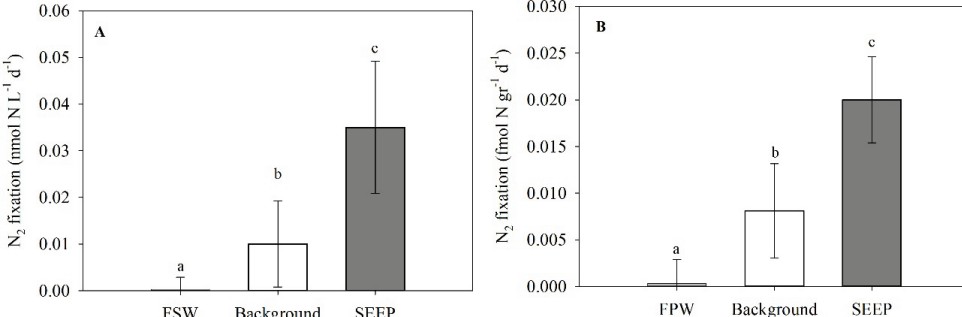

**Figure 9** – $N_2$ fixation rates at the overlying water (A, ~1 m above seabed) and in the top sediment (B, upper 3 cm) in an active SEEP site (AG16-17BC1, grey) and in a background station (AG16-15BC1). The plot shows averages and corresponding standard deviation of three replicates. Filtered seawater (FSW) or filtered porewater (FPW) served as a control.

### 4. Conclusions

Our results indicate that SEMS seep chemotones are characterized by extensive burrowing by ghost shrimp *C. lobata*. The burrowed sediments can extend tens to hundreds of meters away from the seepage hotspots, extending the seep area of influence by at least an order of magnitude. The high microbial activity in these burrowed sediments alters the biogeochemical cycles at the sediment-water interface of the oligotrophic SEMS (Sisma-Ventura et al., 2022), and our new results indicate that unique microbial communities at the burrow walls can contribute new functions needed to sustain this activity. These functions include the fermentation of complex macromolecules by Bacterioida, coupling oxidation of these fermentation products to sulfate reduction by Desulfobacterota and Nitrospirota, and chemotrophy by Campylobacterota and chemosynthetic Gammaproteobacteria, including the giant sulfur bacteria *Thiomargarita*. Enhanced nitrogen cycling at seep chemotones may play a particularly important role in the oligotrophic SEMS.

The functionality of the burrow communities may depend on the gardening or grooming of burrow walls by *C. lobata*. Our observations indicate that this behavior is ubiquitous in ghost shrimps (Axiidea), both in shallow and deep habitats. While the burrow wall communities are most likely fueled by *C. lobata* secretions, producing an organic matter sink ('detrital trap') in the burrow (Papaspyrou et al., 2005), it remains to be tested if *C. lobata* burrows can reach the sulfate-methane transition zone (e.g., the deeper sections of the sampled sediments, where ANME-1 were found, **Figure 3**), providing substrates for chemosynthesis. It is also intriguing to investigate the evolutionary aspects of crustacean associations with microbes in chemosynthetic habitats, given the similarity between the burrow and ectosymbiont communities (Xu et al., 2022; Bai et al., 2021; Cambon-Bonavita et al., 2021).

### Author contributions

MR-B, YM, AF, OB and BH conceived this study and acquired funding. YM and OE performed mapping and led ROV/AUV work. AF and OB performed CT scans. GS-V and BH analyzed the chemical parameters. ER estimated microbial abundance and productivity. ZH and YY performed molecular work. MR-B performed molecular and bioinformatics analyses. MR-B wrote the paper with the contributions of all co-authors.



**Acknowledgments**

This research used samples and data provided by the E/V Nautilus Exploration Program, expedition NA019. The authors thank all individuals who helped during the expeditions, including onboard technical and scientific personnel of E/V Nautilus, SEMSEEPS 2016 and IOLR-University of Haifa cruises, and the captains and crews of E/V Nautilus, IOLR R/V Bat-Galim and HCMR R/V Aegaeo. We thank the University of Haifa Hatter Department of Marine Technologies team,
Ben Herzberg and Samuel Cohen, who operated the Yona ROV. We thank Drs. Moshe Tom and Hadas Lubinevsky (IOLR) for their assistance in sample collection.

**Funding**

This study is funded by the Israeli Science Foundation (ISF) grants 913/19 and 1359/23, the Israel Ministry of Energy grants 221-17-002 and 221-17-004, the Israel Ministry of Science and Technology grant 1126, and the Mediterranean Sea
Research Center of Israel (MERCI). The EUROFLEETS2 SEMSEEP cruise was funded by the European Union FP7 Programme under grant agreement no. 312762 and farther analysis was funded by the Swiss National Science Foundation grant Ref. 200021_175587. This work was partly supported by the National Monitoring Program of Israel's Mediterranean waters and the Charney School of Marine Sciences (CSMS), University of Haifa.

**Declaration of competing interest**

The authors declare that the research was conducted in the absence of any commercial or financial relationships that could be construed as a potential conflict of interest.

**Data availability statement**

The raw metagenomic reads and metagenome-assembled genomes are available on NCBI as BioProject PRJNA1072319.

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
