# Peer review of "Animal burrowing at cold seep ecotones boosts productivity by linking macromolecule turnover with chemosynthesis and nutrient cycling"

_EGUsphere, 2024_

## Author Response (AR1)

Review of egusphere-2024-1285: Animal burrowing at cold seep ecotones boosts productivity by linking macromolecule turnover with chemosynthesis and nutrient cycling

Dear Andrew Thurber,

Thank you and the reviewers for considering this manuscript for publication in Biogeosciences and providing valuable feedback!

Please see our responses below in bold font:

Reviewer 1:

Firstly, it has to be acknowledged that reviewing this MS poses a significant challenge, requiring the reviewer to possess a substantial depth of knowledge encompassing various fields such as seabed geomorphology, geochemistry, microbiology, among others. This consideration was taken into account when I decided to accept the invitation to review the MS. Therefore, my comments can not be treated as a comprehensive assessment but rather focuses on: (1) acquiring relevant knowledge from the MS; and (2) evaluating certain aspects of the MS based on my professional expertise (geochemistry). For the assessment of geomorphology and microbiology, additional reviewers are required to finalize the evaluation.

Fortunately, I now finished the review of this MS. I believe this study provides a detailed description of the biogeochemical processes around the cold seep ecosystem (also referred to as "chemotones" by the author) to some extent. Therefore, it is highly suitable for publication in BG. The data is of good quality and the demonstrated geochemical trends as well as the suggested explanations are convincing. I recommend to accept the paper for publication after a minor revision.

**We thank the reviewer in accepting the challenge of reviewing this manuscript, and the positive feedback!**

An important observation regarding this MS is the extensive discussion presented in sections 3.1 to 3.8, which may potentially divert the attention of readers. It is recommended that the author consider consolidating the discussion content, but I have no clue to achieve this kind of integration to be honest.

**We reviewed these sections, which are typically written to be most concise. We believe that taken together, these parts are needed to tell the story. We agree that some readers will focus only on specific sections.**

**Minor suggestions:**

Line 92: There is no need to give 'SAS' again, since this abbreviation has been given in line 68.

**Thank you for pointing this out, corrected as suggested.**

Table 1: According to the information in the table, it seems that no 'metagenomics' investigation on any samples, why still keep '6. metagenomics' in caption?

**We fixed the table, as there was a glitch in numbering.**

Figure 1: The scale should be optimized. It can be arranged at equal intervals of 0, 5, 10, 15 and 20, and the unit (km) can be put at the end (behind '20').

**Corrected as suggested**

Figure 2: The scale is given above the subgraph D, but the specific length represented by the scale is not specified in the caption. Subgraph E can be enlarged appropriately to make it highly consistent with subgraph D. At the same time, compared with other characters in this figure, the font size of the characters next to subgraph E is too small, which is not friendly to readers.

**Corrected as suggested**

Figure 3: Inconsistencies in the spacing between subgraphs, affecting the overall aesthetic appeal. Specifically, (1) the distance between subgraph B and the adjacent subgraphs is irregular. (2) the proximity between subgraphs C and D is notably smaller compared to the distance among subgraphs A, E, and F, which is considerably larger. (3) the height of subgraph B exceeds that of the combined height of subgraphs A+E+F. (4) subgraph D have two "2 mm" around the scalebar in the lower left corner.

**Corrected as suggested**

Figure 4: The ordinate of subgraph C is incorrect. Or the subgraph B covers a part of the ordinate of the subgraph C. In any case, it needs to be revised. The horizontal and vertical fonts of subgraph G are different from other subgraphs.

**Corrected as suggested**

Figure 6: The correlation of subgraph C is relatively weak. Moreover, the data itself exhibits a considerable error bar, raising uncertainty regarding the appropriateness of discussing the correlation based on this data.

**We agree that this correlation is weak, and now explicitly highlight this in the text. We believe that it still important to present this potential trend, leading to further questions regarding microbial activity in bioturbated sediments. As we noted, the measurements were made on the sections, but not specifically in the vicinity of burrows, likely diluting the signal, which could be more pronounced.**

Figure 9: The figures in this MS display the utilization of various fonts. Specifically, "Times New Roman" is employed in Figure 9, but it seems that this font is not used in other figures. It is recommended that the author adhere to a consistent font style in accordance with the journal's guidelines.

**Corrected as suggested**

Reviewer 2:

The manuscript focuses on the "chemotone," an important but underexplored zone in prior research. By analyzing community composition, activity measurements, and geochemical gradients, the authors suggest that the ecological influence of cold seeps may have been underestimated. This underestimation arises from the overlooked contribution of geochemical recycling within sediments, which is enhanced by burrow effects—particularly in carbon, nitrogen, and sulfur cycling. The methodology is robust, and the reasoning is well-founded. Therefore, I recommend accepting the manuscript with minor revisions.

**We thank Prof. Wang Minxiao for the positive review and valuable remarks!**

Suggestions for Improvement:

Sample Terminology: I strongly suggest the authors use clearer and more intuitive sample descriptions. During my review, I frequently had to return to the methods section table to understand each sample's characteristics (e.g., location, degree of burrowing). Adopting more descriptive and accessible terminology would significantly enhance the manuscript's clarity and readability.

**Thank you for pointing this out. We fixed the sample descriptions throughout the manuscript to better reflect the origin of collection. For example, "Box1" now reads "RC20-BC1", to emphasize the**

Figure Improvements:

Figure 2D: The scale bar is represented only as a line segment without numerical values. Please add specific numerical values for clarity.

**Corrected as suggested**

Figure 3A: The "decimeter-scale mounding and multiple holes and scratches" described in the text are not clearly visible in the figure. I recommend using arrows or labels to better indicate these features.

**We added arrows for holes and scratches. The decimeter-scale mounding is indeed not visible in this image, thus we removed it from the legend.**

Figure 4C: The Y-axis values appear distorted, possibly due to overlapping image layers. This should be corrected to ensure accuracy and clarity.

**Corrected as suggested**

Line 528: The authors state, "Our observations indicate that this behavior is ubiquitous in ghost shrimps (Axiidea), both in shallow and deep habitats." However, the discussion does not address the burrowing behavior of shallow-water ghost shrimps. Including a brief comparison of shallow and deep burrowing behaviors would provide a more comprehensive perspective and strengthen the discussion.

**We modified this text and added a reference to burrowing by shallow-water shrimps (Laverocket al. 2010, Bioturbating shrimp alter the structure and diversity of bacterial communities in coastal marine sediments, doi:10.1038/ismej.2010.86 and Papaspyrouet al 2005, Sediment properties and bacterial community in burrows of the ghost shrimp *Pestarella tyrrhena* (Decapoda: Thalassinidea), doi:10.3354/ame038181)**
***"Our observations indicate that this behavior is ubiquitous in ghost shrimps, altering benthic-pelagic nutrient exchange in both shallow and deep habitats (Laverock et al., 2010; Papaspyrou et al., 2005)."***

**Introduction and Discussion**:

The introduction devotes considerable attention to the "chemotone." While this highlights its relevance, I believe the authors intend to emphasize the underappreciated role of this zone in ecosystem dynamics. In the discussion, I suggest briefly underscoring that the influence of cold seeps has likely been underestimated in previous studies, further reinforcing the importance of the findings.

**As suggested, we added the following text in the conclusions: "The discovery of these functions underscores the underestimated role of seep chemotones in deep-sea biogeochemistry."**

I wish to see your future work to show the vertical depth of the shrimp can reach as the methane oxidation is generally limited by the availability of the oxygen, nitrate or sulfate.

**Thank you for this comment! We agree that future detailed investigation is needed to understand how shrimp burrowing is linked to methane fluxes in the seep habitat.**

---

## Author Response (AR2)

Review of egusphere-2024-1285: Animal burrowing at cold seep ecotones boosts productivity by linking macromolecule turnover with chemosynthesis and nutrient cycling

Dear Andrew Thurber,

Thank you for the feedback and for considering the manuscript for publication!

We made an additional attempt to streamline sections 3.3-3.8. For that:

1) We moved the discussion of N2 fixation into section 3.3  - burrowing alters geochemical processes at seep chemotones. We also placed Figure 9 in the Supplement, to reduce the load in the main text.
2) We streamlined the first paragraph of section 3.6 (description of diversity based on metagenome-assembled genomes) and moved it to the end of section 3.5.
3) Thoroughly revised section 3.6, condensing the content.

We hope that these changes satisfactory consolidate the messaging of the section.

Maxim